# Source Identification and Quantification of Seepage Water in a Coastal Mine, in China

**Xueliang Duan** [1,2,3]**, Fengshan Ma** [1,2,]*****, Jie Guo** [1,2]**, Haijun Zhao** [1,2]**, Hongyu Gu** [4]**, Shuaiqi Liu** [1,2,3] **and Qihao Sun** [1,2,3]

[1]   Key Laboratory of Shale Gas and Geoengineering, Institute of Geology and Geophysics, Chinese Academy of Sciences, Beijing 100029, China
[2]   Institutions of Earth Science, Chinese Academy of Sciences, Beijing 100029, China
[3]   University of Chinese Academy of Sciences, Beijing 100049, China
[4]   Chengdu Center, China Geological Survey, Chengdu 610081, China
*****   Correspondence: fsma@mail.iggcas.ac.cn

**Abstract:** The Sanshandao gold mine, which is the largest coastal mine in China, is under threat from seawater intrusion and water inrush. The objective of this study is to determine the water end-members (seawater, freshwater, and brine) of the seepage water in the mine and quantify the proportion of end-members. Non-conservative ions and ion exchange were identified by using hydrogeochemical analysis. Then, the principal component analysis (PCA) was used to identify the end-members of mine water. Three end-members were identified, so a ternary mixture model was applied to compute the mixing ratios. The potential water flow channels and the prevailing supply patterns were inferred by combining the results of mixing ratios with the tectonic and engineering geological conditions. The results indicate that the proportion of seawater in mine water is about 57%, the freshwater is about 16% and the brine is about 27% for the entire mine area, the prevailing supply pattern of seawater was lateral recharge, the water samples which were located in −510 m sublevel or in the northeast of prospecting line 2260 had high proportions of seawater, the freshwater supplied the groundwater mainly through the secondary fractures developed area in a vertical recharge and the influence depth was about −500 m, and F3 was the largest tensile-shear fault in the study area and it was both a watercourse for seawater and fresh water.

**Keywords:** coastal gold mine; hydrogeochemical analysis; principal component analysis; mixing ratios

## 1. Introduction

After years of mining, the limited land resources have been depleted, so mining is gradually transferring to coastal and undersea resources [1–5]. Especially in China, coastal gold mining is becoming a major objective of the gold mining industry [6]. Our study area, the Sanshandao gold mine, is the largest coastal gold mine in China. Because of its special geographical position, mining operations are mostly concentrated below sea level. Thus, seawater is a potential threat to the mine, and the hydraulic connectivity of mine water and sea water should be the focus of attention [7–9]. Thus, it is necessary to identify the water end-members and calculate the mixing ratios in order to provide support and guarantee the safety of deep coastal mining activities.

Hydrogeochemical and stable isotope analyses have been widely used to identify water from different origins [10]. Guo et al. (2015) [11] studied the origin of water and calculated the mixing ratios in a coastal mine by using hydrochemical element factors and stable isotopes ($\delta^2$H and $\delta^{18}$O). Li et al. (2016) [12] estimated the hydraulic connection between ground water and surface water

according to analyses of hydro-chemistry and isotopic signature. Liu et al. (2007) [13] determined the sources of recharge proportionally based on $^{18}$O isotopic and water chemistry data measured for each drainage point in the pits. Fan et al. (2016) [14] identified the water sources of a river by using isotopic ratios. The use of only conservative species in this method is more reliable and can avoid errors produced by non-conservative species [15], and there are still many conservative species which are not used for analysis, which means only a small part of the water sample information is considered. Thus, the analysis result is not comprehensive. Principal component analysis (PCA) is a multivariate statistic method which is useful in data reduction, manipulation, and visualization of complex data systems [16–19]. PCA can transform a multivariate data set into several principal components which are linear combinations of the original variables. In general, the new principal components can account for the most information of original variables [20,21]. Therefore, using principal components is more representative than using some of the original variables in the calculation of mixing ratios. Thus, the second purpose of using PCA was to get more precise calculation results of mixing ratios. However, if non-conservative species are included in the analysis, the result of the PCA would be poor. In order to overcome the above problems, this study combined hydrogeochemical analysis with PCA. First, hydrogeochemical analysis was used to identify the non-conservative ions. Then, the water end-members were analyzed by using PCA.

It is necessary to identify the water sources and it is also important to quantify the end-members in the mixed water. Some calculation methods of mixing ratio, such as MIX (a maximum likelihood method to estimate mixing ratios) and UMMIX (a multivariate receptor model) [22,23], were designed to achieve mathematically optimal solutions so sometimes they could produce results that do not fit the facts, for example, the concentration of end-member is negative. The mixing model based on mass balance of chemical and isotopic species is often used in mixing calculation [24,25], and the selected species should be conservative so as to get precise calculation results. However, usually one or two chemical and isotopic species can not represent most of the information of the water sample. Thus, combining the PCA with this mixing model can be more effective and the chemical and isotopic species in the original mixing model can be replaced by the principle components.

The Sanshandao gold deposit is a typical tectonically fractured, water-filled mine. The groundwater dynamics (the storage and transport of groundwater) are mainly controlled by tectonic structures, including faults and cracks. Therefore, in deep underground granite, fracture flow dominates while the matrix flow is mainly present in the Quaternary aquifer in the shallow ground. Thus, fissures (underground watercourse) can be inferred by the seepage water in the tunnel of the mine. The underground flow pattern can be inferred by the results of mixing ratios combined with the tectonic and engineering geological conditions of the study area. Under the influence of mining activities, the flow rate of groundwater in the mine is high. Because of the high flow rate and low temperature of mine water, the chemical and isotopic reactions (ion exchange and water–rock interaction) have little effect on it. Thus, the mine water is mainly dominated by the processes of mixing. The ion exchange reactions were identified by hydrogeochemical analysis and the isotopic reactions were analyzed in the deviation analysis of the mixing ratios.

Thus, there are two aims of this paper: (i) to identify and quantify the water sources of the mine water in Sanshandao gold mine; and (ii) to analyze the potential water flow channels and the prevailing supply patterns. In summary, we hope that the research methods and results will provide insight for water management and help inform models used to design preventative measures of water inrush in coastal mines which are similar to the Sanshandao gold deposit.

## 2. Study Area

The Sanshandao gold deposit is situated in the north of Laizhou City, bordering Bohai Sea in the northwest and connected to the land in the southeast (Figure 1). There are three hills near the sea in the north of the study area. The rest is flat terrain, with a ground elevation of 1 to 6 m.

## 2.1. Geological Setting

The study area is located 20 km to the east of the Tan-Lu fault, the north-limb of the Qixia anticlinorium, and the northeastern part of the Sanshandao–Cangshang fault zone.

The stratum in this study area includes Quaternary strata and Archean Jiaodong Group strata. Quaternary strata are widely distributed in the mining area (except for the three hills), and the thickness gradually increases from the foot of the three hills to the southeast, with a maximum thickness of about 50 m. From top to bottom, the lithology is: medium and coarse sand layer; sandy clay and clay sand layer; medium, coarse sand, and gravel layer; gravel sandy clay and clay layer. Jiaodong Group strata is composed of amphibolites and biotite gneiss. It is distributed on the southeast side of the mining area and is covered by the Quaternary sediments. The Yanshanian biotite granite (the Linglong granite in the area) is developed on the northwest side of the mining area. There are three faults with large control significance in the mining area, namely F1, F2, and F3 [26,27].

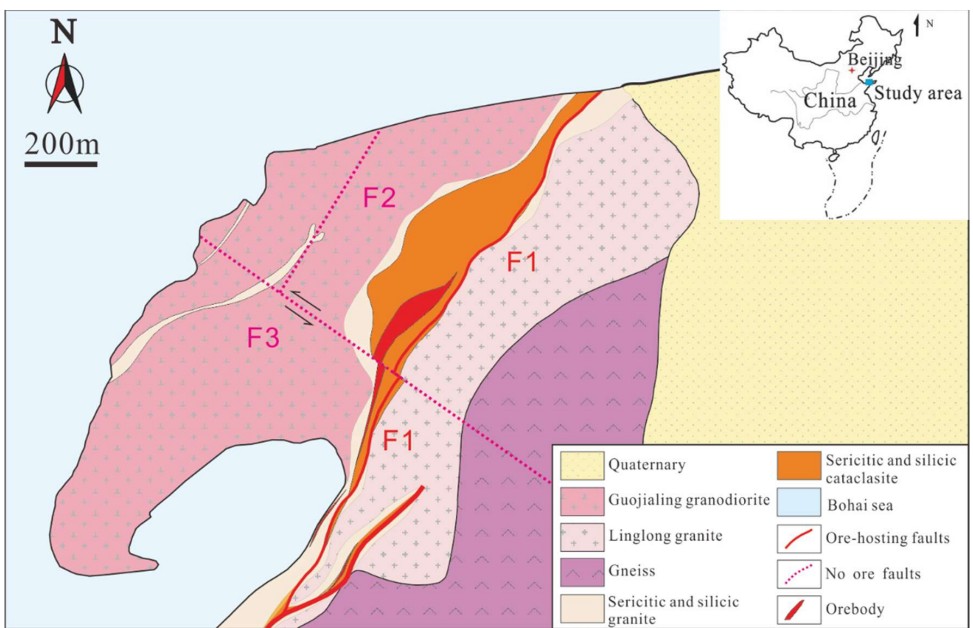

**Figure 1.** Geological map of the Sanshandao gold mine.

The Sanshandao–Cangshang fault zone (ore-forming structure) is 11 km long and 40–250 m wide. It enters the Bohai Sea at the north end and extends into Laizhou Bay to the south. The fault zone is developed in the granite of the contact zone between Linglong granite and Jiaodong Group.

F1 (a compresso-shearing fault) is a part of the Sanshandao–Cangshang fault zone and the Sanshandao gold deposit is controlled by F1. It formed before the gold mineralization and its attitude is generally NE 35–40°/SE ∠40°, with undulating characteristics along the strike and inclination. The hanging wall of F1 consists of beresitized chip granite and biotite granite. The foot wall is composed mainly of beresitization mylonite that is mineralized. The gold ore body is distributed in the foot wall of F1. A continuous layer of black fault gouge is developed on the major fracture plane, which is about 1–10 cm thick. Under normal conditions, the fault gouge is tightly squeezed, which excludes water.

F2 is a tense-shearing fault with an attitude of NE15°/NW∠60–80°. Its northern end extends into the Bohai Sea, and the southern end intersects with F3. The rupture behavior of rock mass along the strike changes greatly, and the water conductivity is segmental.

F3 (a tense-shearing fault) is located in the southern region of the mining area, near the 1780 prospecting line (Figure 2) and extends to the Bohai Sea in the northwest. It is the largest NNW fault in the mining area. The average occurrence is NW305°/NE∠85°. Its fractured zone is 5–30 m in width

and filled with mafic dikes, including diabase and lamprophyre. F3 is a large regional fault and has undergone several tectonic movements, especially the late stage activities. This makes the rock in the fracture zone extremely fractured, which provides space for the storage and migration of groundwater. F3 offsets the ore body and F1 by 10 to 20 m, which makes the aquifuge of F1 partially destroyed (the fault gouge of F1 is cut). In addition, F3 connects the mining area with sea water.

## 2.2. Hydrogeological Conditions

According to the spatial location, the groundwater system in the mining area can be divided into three sub-systems: the overlying Quaternary pore water, the bed bedrock fissure water in the hanging wall of F1, and the bed bedrock fissure water in the foot wall of F1.

The Quaternary aquifer is distributed throughout the upper part of the mine and is recharged by freshwater and seawater. The permeability coefficient of the aquifer is 1.9–117.5 m/d. The aquifuge at the bottom of the Quaternary is composed of gravel sandy clay and clay. It is located on the bedrock weathering crust. The average buried depth is 25.5 m, the thickness is generally 3 to 5 m, and the thickest is 19.6 m. It has low porosity and permeability so its water-insulating capability is strong. Generally, there is no hydraulic connection between the Quaternary aquifer and lower bedrock fissures. However, due to the influence of the Quaternary active fault, the bedrock fissure groundwater will be replenished in the local area.

The aquifer system of hanging wall consists of weathering fissures developed in the granite of hanging wall of F1. Because the floor (hard and compact biotite granite) of the aquifer and the gouge of F1 both exclude water, there is no hydraulic connection between hanging and foot wall, except for the contact zone with F3.

Some areas in the foot wall of F1 have become water-rich areas due to the development of NW striking fault and fracture zones and the joint fissures of the NE direction, and have hydraulic connections with the northern and western seawater. There are mainly two water-rich areas: (1) F3 and its influence zone. The water-conducting and storage space are formed by F3 and extend to the BoHai Sea to the northwest. (2) The "fracture zone" is distributed between the prospecting line 1780 and 2260 (Figure 2). A series of secondary fractures (NW striking small fault and NW and NE striking joint fissures) are developed in this area. These are tensile fractures and they have large apertures which provide space for water migration and storage [28–30].

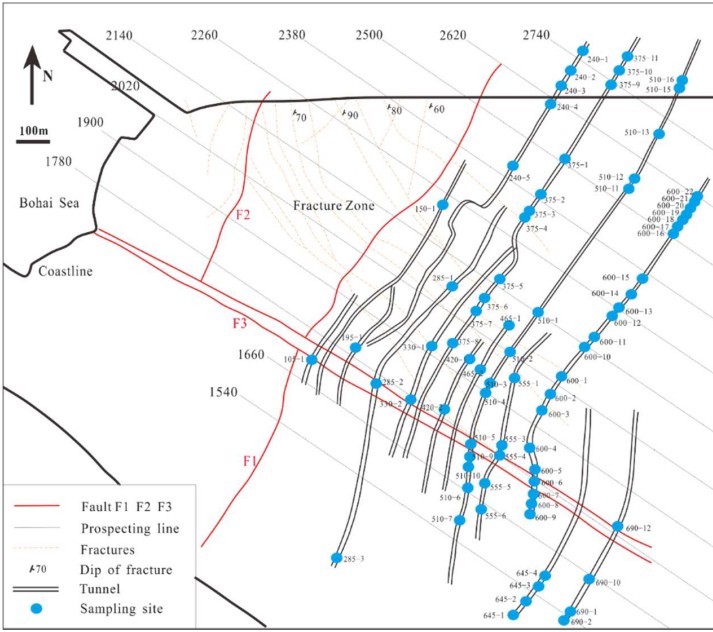

**Figure 2.** Location of water samples.

## 3. Materials and Methods

### 3.1. Sampling and Analytical Procedures

The water samples to be collected include two kinds, one is the seepage water in the mine and the other are four water sources: Quaternary aquifer water, brine, seawater, and freshwater. The Quaternary aquifer water refers to water hosted in the Quaternary aquifer and it is recharged by fresh water and sea water. For the convenience of narration, the Quaternary aquifer water is simply referred to as Quaternary water. Brine refers to water stored in underground bedrock fissures. The brine has a high concentration of total dissolved solids and is considered to originate from evaporating and concentrating of paleo-seawater [10,31,32]. Seawater refers to water in the Bohai Sea located northwest of the mining area. Freshwater refers to the surface water in the Wanghe River located south of the mining area.

The seepage water samples were collected periodically in sublevels of the mine in August every year. From 2009 to 2015 (except for 2010), six campaigns of water sampling were performed in the mining area and 165 samples were collected in 14 sublevels (from −105 m sublevel to −690 m sublevel at intervals of 45 m). Cracks in the mine roadway with water flow were set to sampling sites. The name of the sampling site consists of two parts. The first half represents the sublevel name and the second half represents the numbering of the sampling site in this sublevel. For example, the No. 2 sampling site in the −600 m sublevel is named as 600-2 (Figure 2).

In 2009, 2 seawater samples, 4 freshwater samples, and 5 Quaternary water samples were collected. Seawater was pumped 1 m below sea level in the BoHai Sea. Freshwater was obtained from the Wanghe River. Quaternary water was collected from the Quaternary aquifer (at a depth of 30 m) by drilling south of the mining area. The brine was sampled by drilling in the tunnel of −375 m sublevel in 2015.

At each sampling site, two identical bottles (each bottle had a capacity of 600 mL) were used for collecting water samples, one was filled for testing water chemistry and the other for testing isotope. For quality control, the water samples were preserved at 5 °C in brown polyethylene (PE) plastic bottles with air-tight seals prior to analysis. The water chemical analysis was carried out in the Institute of Geology, China earthquake administration. Parameters that needed to be analyzed included major ions ($K^+$, $Na^+$, $Ca^{2+}$, $Mg^{2+}$, $Cl^-$, $SO_4^{2-}$, $HCO_3^-$, and $NO_3^-$), pH, electrical conductivity (EC), total dissolved solids (TDS), and total hardness (TH). The DIONEX-500 ion chromatograph (Manufactured by Thermo Fisher Scientific, Massachusetts, MA, USA) was used to analyze water samples. The accuracy of cations, anions, pH, EC, and TDS were 0.01 mg/L, 0.1 mg/L, 0.01, 0.1 μS/cm, and 0.01 mg/L, respectively. All water samples were analyzed based on the test method and the following test standards: DZ/T0064.28-1993, DZ/T0064.29-1993, and DZ/T0064.51-1993. The isotope ($\delta^{18}O$ and $\delta D$) analysis was performed at the laboratory for stable isotope geochemistry, institute of geology and geophysics, Chinese academy of sciences. The hydrogen-isotope-ratio and oxygen-isotope-ratio were analyzed using the MT-253 mass spectrometer. The isotope ratios are expressed in per mil (‰) relative to VSMOW (Vienna Standard Mean Ocean Water). The analytical precision of $\delta^{18}O$ and $\delta D$ were 0.2‰ and 2‰, respectively [15].

### 3.2. Principal Component Analysis (Based on Hydrogeochemical Analysis)

The PCA results are robust on the premise that the species analyzed are conservative. Thus, the hydrogeochemical analysis was used to select the non-conservative ions. On this basis, the conservative species were analyzed by PCA to identify water end-members. The calculation process of PCA has been standardized. However, the number of principal components (PCs) to be retained should be artificially determined. Most researchers use the Kaiser criterion (all PCs with eigenvalues greater than one) to determine the number of PCs. But this criterion does not always produce the best results [33]. The explained variance of the PCs can also be used to select PCs [34]. In general, the variance of the PCs should amount to 70%. In this study, both of the two methods were used to determine PCs.

## 3.3. Mixing Calculation

Water balances is often evaluated by using the method of mass balance [22]. The mixing calculation is based on the law of mass conservation. The simplest mixing pattern is a binary mixing model. An ideal binary mixing model is shown as the Equations (1) and (2). A N-components model is developed from the binary model and it can be written in matrix form (Equation (3)). In the actual calculation, the N is determined by the number of the end-members.

$$\delta_1 a_1 + \delta_2 a_2 = a_{sp} \tag{1}$$

$$\delta_1 + \delta_2 = 1 \tag{2}$$

where $a_1$ and $a_2$ are the concentrations of species $a$ of end-member 1 and end-member 2, respectively. $a_{sp}$ is the concentration of species a of the $p$th mine water sample. $\delta_1$ and $\delta_2$ are the mixing ratios of the $p$th mine water sample.

$$\begin{bmatrix} a_1 & a_2 & \cdots & a_n \\ b_1 & b_2 & \cdots & b_n \\ \vdots & \vdots & \vdots & \vdots \\ 1 & 1 & \cdots & 1 \end{bmatrix} \begin{bmatrix} \delta_{1p} \\ \delta_{2p} \\ \vdots \\ \delta_{np} \end{bmatrix} = \begin{bmatrix} a_{sp} \\ b_{sp} \\ \vdots \\ 1 \end{bmatrix} \tag{3}$$

where $[a_1, a_2, \ldots a_n]$ are the concentrations of species $a$ of end-member 1, end-member 2, $\ldots$ and end-member n, respectively. $[b_1, b_2, \ldots b_n]$ are the concentrations of species $b$ of end-member 1, end-member 2, $\ldots$ and end-member $n$, respectively. $[\delta_1, \delta_2, \ldots \delta_n]$ are the mixing ratios of the $p$th mine water sample. $a_{sp}$ and $b_{sp}$ are the concentrations of species $a$ and species $b$ of the $p$th mine water sample.

## 3.4. Deviation Analysis

The mixing model mentioned above is in the ideal condition, while in a practical situation there is no perfect mixture. Take binary mixing model as an example to illustrate (Figure 3). There are two end-members (end-member 1 and end-member 2) and a and b are two species of these. The ideal mixed water samples lie on the mixing line (such as the yellow point in Figure 3), however it a fact that most of the samples do not happen to lie on the line, but lie near the line (such as the red point in Figure 3). Only one species (species a was selected in Figure 3) is selected to calculate in the binary model. For this species, the results of mixing ratios ($\delta_{a1}$ and $\delta_{a2}$ in Figure 3) are accurate. However, in addition to the selected species, there are many other species (such as species b in Figure 3) which are not used in the mixing calculation. Thus, these species will inevitably deviate from the results of mixing ratios. The deviation can be represented by comparing the measured and computed concentrations of each species of mixed water samples. The measured concentrations (concentration $b_m$ in Figure 3) are actually the measured values of the mixed water samples. The computed concentrations (concentration $b_c$ in Figure 3) could be obtained through back calculation by using the end-members concentrations and the result of mixing ratios. The deviation of species b can be expressed as $\Delta b = b_c - b_m$. Thus, generally, the deviation can be quantified by Equation (4).

$$D = \frac{C_c - C_m}{C_m} \tag{4}$$

where $C_c$ is the computed concentration of species (ions or isotopes) and $C_m$ is the measured concentration.

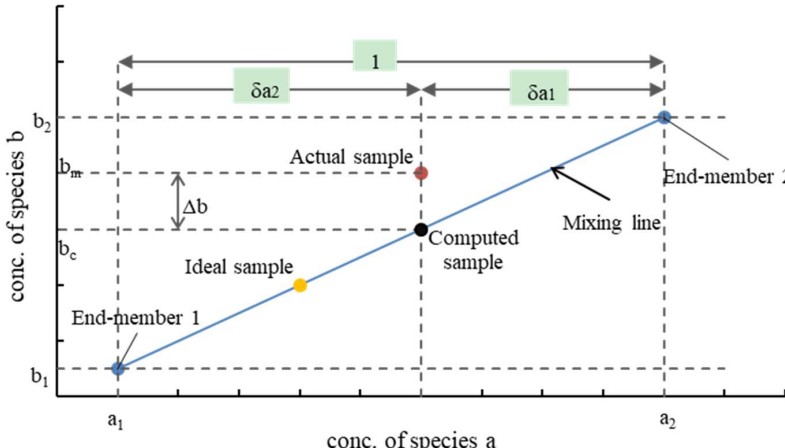

**Figure 3.** The schematic diagram of the binary mixing model. (*x*-axis: concentration of species **a**, *y*-axis: concentration of species **b**).

## 4. Results and Discussion

### 4.1. Hydrochemical Analysis

The hydrochemical analyses can present the characteristics of different water samples. Non-conservative ions and ion exchange can also be identified by using the hydrogeochemical analysis.

The testing results of each water index, including isotope concentrations ($\delta^{18}O$, $\delta D$), major ion concentrations, electrical conductivity (EC), total dissolved solids (TDS), and total hardness (TH) are listed in Table 1. The results include the minimum, maximum, and mean values of each index concentration of 165 mine water samples and the concentrations of four potential water sources (seawater, brine, Quaternary water, and fresh water). The seawater had the highest isotope ratios in all water samples. The concentration of each indicator in freshwater was low. The Quaternary water was supplied by freshwater and seawater, so its concentration was between seawater and freshwater. The brine (375-20) had the highest concentrations of ions, EC, TH, and TDS. Most of the mine waters were highly mineralized water (TDSs in mine water are higher than that in sea water) but the $K^+$ concentrations in mine water were lower than that in sea water. It indicates that rock and soil can adsorb $K^+$ from water [35]. The large floating range of the minimum and maximum values of the $K^+$ concentrations of mine water indicates that the $K^+$ is a non-conservative species.

$Cl^-$ is a conservative ion because it is not adsorbed by rock and soil, is not absorbed by bacteria and plants, has high solubility and is not easy to precipitate out before halite saturation is reached [36]. Thus, the characteristics of other ions can be analyzed by comparing the relationship with chloride ion. There were significant positive correlations between chloride ion and most ions including $Na^+$, $SO_4^{2-}$, EC, TDS, and TH. Take $Na^+$ as an example (Figure 4a).

The correlation index is $R^2 = 0.926$ (the closer the $R^2$ is to 1, the better the correlation). While there were not linear correlations between $Cl^-$ and $K^+$, $Ca^{2+}$ and $Mg^{2+}$, the distribution of $K^+$ was discrete and irregular because it was a non-conservative species. The correlations between $Cl^-$, $Ca^{2+}$, and $Mg^{2+}$ are shown in Figure 4 (Figure 4b,c show the data for 2011, and the water samples for other years have the same pattern). For most water samples, $Cl^-$ had good linear correlations with $Ca^{2+}$ and $Mg^{2+}$, except for the outliers (the red points in the figure). In Figure 4b, the outliers had a high concentration of $Ca^{2+}$, and in Figure 4c, the outliers had a low concentration of $Mg^{2+}$. The outliers in two figures were almost the same water samples which are collected in the "fracture zone" (Figure 2). Thus, when the water flowed through or stored in these cracks, the $Ca^{2+}$ on the surface of the soil is interpreted to exchange the $Mg^{2+}$ in the mixed water, which led to the increase of $Ca^{2+}$ and the decrease of $Mg^{2+}$ in the mine water [37]. In addition, the effect of carbonate dissolution on the concentration of $Ca^{2+}$ and $Mg^{2+}$ was also analyzed (Figure 4e,f). $Ca^{2+}$ are negatively correlated with $HCO_3^-$, while $Mg^{2+}$ are positively correlated with $HCO_3^-$. Therefore, this can also be regarded as the exchange of $Ca^{2+}$ and

$Mg^{2+}$. That is, the effect of $HCO_3^-$ on $Ca^{2+} + Mg^{2+}$ is small. Although there was the ion exchange between $Ca^{2+}$ and $Mg^{2+}$, the total amount of $Ca^{2+} + Mg^{2+}$ is constant. The correlation between $Cl^-$ and $Ca^{2+} + Mg^{2+}$ is shown in Figure 4d, and the fit line is produced by water samples in 2011, except 285-3. There was a good linear correlation ($R^2 = 0.912$) between $Cl^-$ and $Ca^{2+} + Mg^{2+}$ (Figure 4d). Thus, the $Ca^{2+} + Mg^{2+}$ should be analyzed as a whole.

**Table 1.** The analyses results of mine water and four potential water sources, including isotope (‰), concentrations of ions (mg/L), EC (S/cm), TDS (mg/L), and TH (mg/L).

| | Mine Water | | | Seawater | Brine (375-20) | Freshwater | Quaternary Water |
|---|---|---|---|---|---|---|---|
| | Minimum | Maximum | Mean | | | | |
| $\delta^{18}O$ | −5.60 | −1.05 | −2.17 | −0.18 | −3.09 | −8.03 | −2.36 |
| $\delta D$ | −43.25 | −2.83 | −17.85 | −5.18 | −21.60 | −57.82 | −24.00 |
| $K^+$ | 35.50 | 370.00 | 234.11 | 362.50 | 220.00 | 11.80 | 362.50 |
| $Na^+$ | 6225.00 | 18,062.50 | 10,585.96 | 9050.00 | 20,000.00 | 188.20 | 9750.00 |
| $Ca^{2+}$ | 424.80 | 5090.20 | 1203.13 | 380.80 | 2116.20 | 138.30 | 320.60 |
| $Mg^{2+}$ | 194.40 | 2259.90 | 1184.79 | 1125.10 | 3030.20 | 42.50 | 1093.50 |
| $Cl^-$ | 12,028.20 | 36,544.00 | 20,307.64 | 16,468.30 | 40,196.80 | 212.30 | 17,456.30 |
| $SO_4^{2-}$ | 1306.40 | 3554.20 | 2426.91 | 2439.90 | 3842.40 | 405.40 | 2228.60 |
| EC | 28,700.00 | 63,800.00 | 42,921.21 | 44,400.00 | 64,700.00 | 1760.00 | 46,100.00 |
| TH | 4833.90 | 16,568.20 | 7854.18 | 5584.50 | 17,764.20 | 520.40 | 5304.20 |
| TDS | 22,020.10 | 62,017.80 | 36,131.19 | 29,985.20 | 69,666.10 | 1261.20 | 31,388.40 |

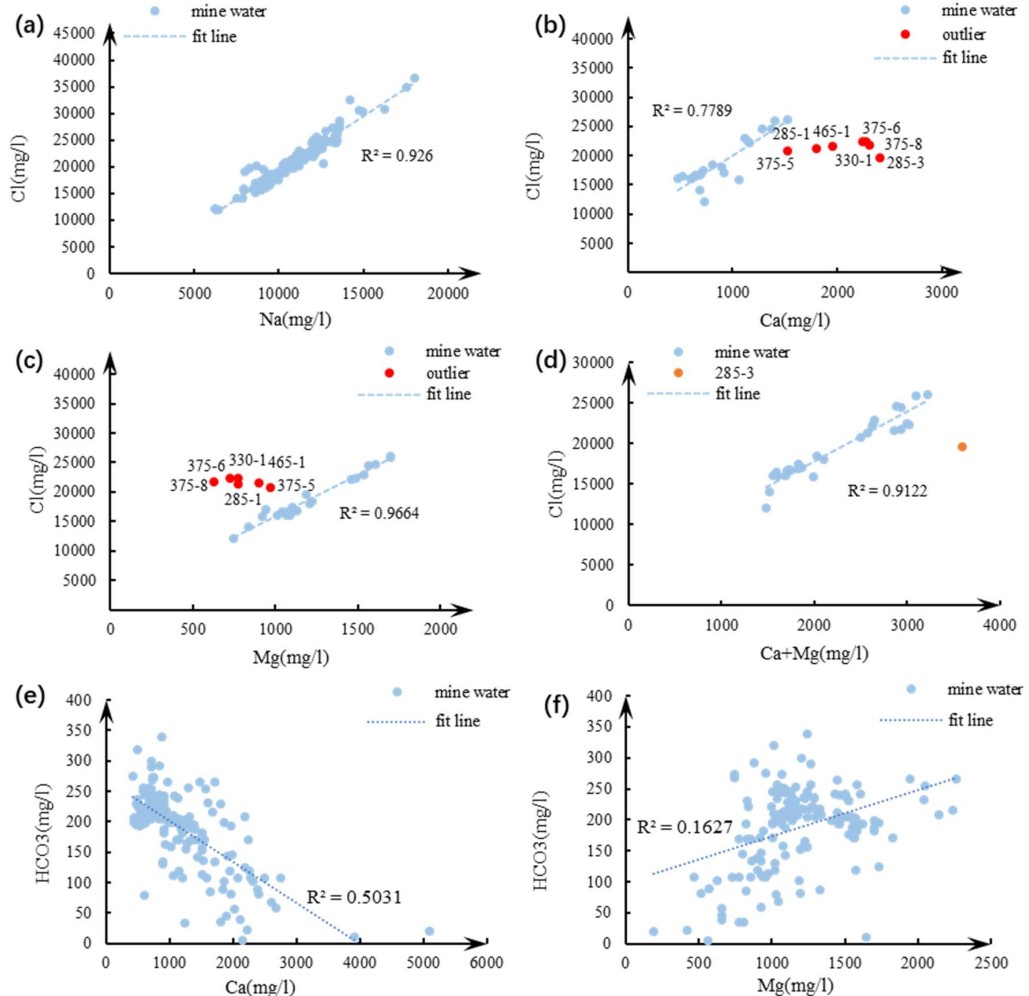

**Figure 4.** Correlation of ions. (**a**) Na vs. Cl, (**b**) Ca vs. Cl, (**c**) Mg vs. Cl, (**d**) Ca + Mg vs. Cl, (**e**) Ca vs. $HCO_3$, and (**f**) Mg vs $HCO_3$.

*4.2. Principal Component Analysis (Determination of End-Members)*

Considering the non-conservative ions and ion exchange, $Na^+$, $Cl^-$, $SO_4^{2-}$, $Ca^{2+} + Mg^{2+}$, EC, TH, and TDS were selected to be analyzed in PCA based on the results of hydrochemical analysis, as well as the stable isotopes ($\delta^{18}O$, $\delta D$) also being a relatively conservative species (the small part of isotopic reactions will be analyzed in the deviation analysis) which can be used to identify different water sources. Therefore, the above nine indicators were analyzed in PCA. The main calculated results of principal component analysis are listed in Table 2. The first two principal components were retained according to eigenvalues and explained variance for two reasons [17]. First, according to the Kaiser criterion that the eigenvalue is greater than 1, only the first two PCs meet the requirement. Second, the explained variances of the first two PCs are 72% and 19%, respectively, and the cumulative variance of these is more than 90%, which is usually considered to be large enough [21].

**Table 2.** Component loadings by PCA (principal component analysis) for all water samples from the Sanshandao deposit ($n = 177$).

| Variables | Principal Components | |
|:---:|:---:|:---:|
| | PC1 | PC2 |
| $\delta^{18}O$ | 0.52 | **0.80** |
| $\delta D$ | 0.45 | **0.81** |
| $Na^+$ | **0.98** | −0.01 |
| $Ca^{2+}+Ma^{2+}$ | **0.86** | −0.46 |
| $Cl^-$ | **0.98** | −0.14 |
| $SO_4^{2-}$ | **0.82** | 0.24 |
| EC | **0.94** | −0.01 |
| TH | **0.91** | −0.36 |
| TDS | **0.99** | −0.11 |
| Eigenvalue | 6.48 | 1.72 |
| Explained variance (%) | 71.95 | 19.15 |
| Cumulative % of variance | 71.95 | 91.11 |
| Significant loadings are in bold | | |

The characteristics of all water samples (including four water sources and all mine waters) are obvious, shown in the plot of the relationship between pc1 and pc2 (Figure 5). Three water end-members (seawater, brine, and freshwater) were identified, which are the extreme values in the plot. The Quaternary water was not regarded as an end-member because it was recharged by seawater and freshwater. Almost all mine water samples were located in the triangle formed by the end-members. In addition, some individual mine water samples (285-3, 600-11, 320-8, and 320-9) fell outside the triangle and are marked in red in the plot. 285-3 (2009, 2011, 2012, and 2013) samples lie close to the mixing line of brine and fresh water. This suggests that site 285-3 is less affected by seawater and has a weak hydraulic connection with other water sites. 320-8, 320-9, and 600-11 samples in 2014 were similar to brine because these samples had high TDS. It suggests that they are less affected by other water. Thus, brine is the main component in these samples. In general, the selection of end-members is reasonable.

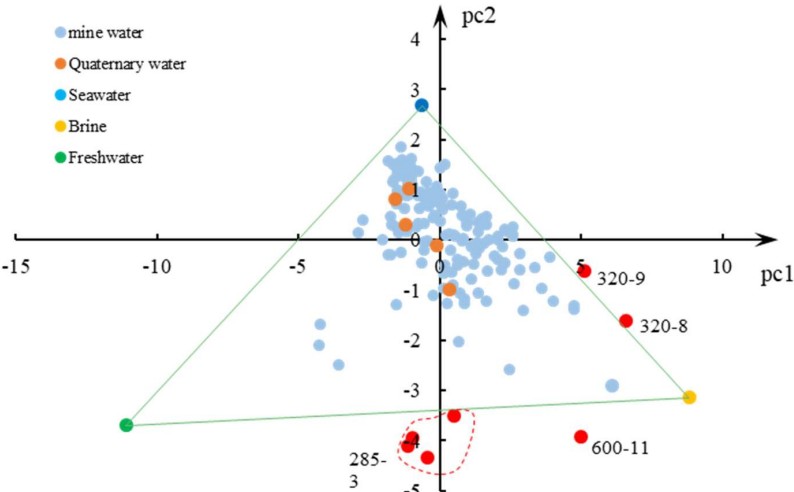

**Figure 5.** Relationship between pc1 (principal component 1) and pc2 of all water samples.

*4.3. Mixing Calculation and Deviation Analysis*

Three water end-members were identified by PCA, so a ternary mixture model can be used to calculate the mixing ratios of the mine water. The principal components, pc1 and pc2, were selected as the calculation species of the mixture model. In general, the proportion of seawater in mine water was about 57%, the freshwater was about 16%, and the brine was about 27% (Figure 6). In −510 m sublevel, the proportions of seawater of the water samples was high, especially at 510-2 (the highest proportion reached 87% in 2011). In addition, 240-4, 285-2, 375-3, 4, 9, 555-4, 600-6, and 600-19 water samples also had high proportions of seawater. These water sites can be divided into two types according to the distribution location. Some were distributed near the northern coast and the others were located near the F3 (Figure 2).

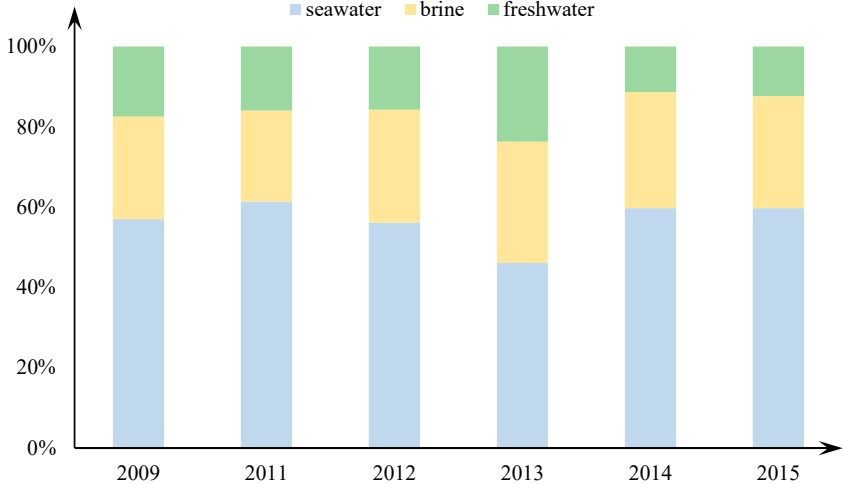

**Figure 6.** Average mixing proportions of all water samples for the entire mine from 2009 to 2015.

The fresh water mainly infiltrated to relatively shallow depths. The water sites with high proportions of freshwater were mainly distributed in the −465 m sublevel and above and between the prospecting line 1780 and 2260 (Figure 2). 150-1 and 285-3 had the highest proportions of freshwater (the proportion of freshwater of these two sites reached 50%).

The deviations were quantified by Equation (4), and the average and maximum deviations of the major ions and isotopes are listed in Table 3. The deviations can also be expressed more intuitively through the relation graph of the measured and calculated concentrations of the water samples which are shown in Figure 7 (the closer the points are to the reference line: y = x, the smaller the deviations,

conversely the greater the deviations). The deviations of stable major ions and isotopes were analyzed. Because there was a strong correlation between $Cl^-$ and EC, and TDS and TH, the analysis results of $Cl^-$ can be used to represent the deviation of other indicators.

**Table 3.** Deviations of all ions and isotopes in two mixing models.

| Model | Deviation | $\delta^{18}O$ | $\delta D$ | $Na^+$ | $Ca^{2+}$ | $Mg^{2+}$ | $Cl^-$ | $SO_4^{2-}$ | $Ca^{2+} + Mg^{2+}$ |
|---|---|---|---|---|---|---|---|---|---|
| | Max (Cc > Cm) | 1.25 | 4.76 | 0.34 | 0.49 | 0.62 | 0.24 | 0.35 | |
| Only PCA | Max (Cc < Cm) | −0.26 | −0.45 | −0.09 | −0.44 | −0.66 | −0.08 | −0.45 | |
| | Mean | 0.19 | 0.26 | 0.00 | −0.11 | 0.11 | 0.01 | −0.03 | |
| PCA (based on | Max (Cc > Cm) | 0.69 | 2.65 | 0.30 | | | 0.22 | 0.78 | 0.15 |
| hydrogeochemical | Max (Cc < Cm) | −0.43 | −0.25 | −0.09 | | | −0.08 | −0.32 | −0.23 |
| analysis) | Mean | 0.04 | 0.11 | 0.00 | | | 0.00 | 0.03 | −0.04 |

For most water samples, there were linear correlations (y = x) between calculated and measured concentrations of each indicators (Figure 7). The linear correlations were obvious, especially for indicators $Na^+$, $Ca^{2+} + Mg^{2+}$, and $Cl^-$. Some points deviated the linear correlation, which are encircled in Figure 7. The 285-3 samples had relatively big deviations. Combined with the previous analysis, the 285-3 site had a weak hydraulic connection with other sites. In addition to 285-3 samples, the 600-11e sample also had big deviations for $Ca^{2+} + Mg^{2+}$, and $SO_4^{2-}$ because it was mostly the original brine with little having affected it other than water. The samples in 2011 and 2014 years had deviations for $\delta D$ (Figure 7b). The slopes of the fit lines of these two years were close to 1, while the intercepts were not equal to 0 (with a small deviation). Thus, there were D-shifts (the enrichment of D) in the water samples. When the space where water is stored is closed for a long time, desulfurization will occur, and $SO_4^{2-}$ will be reduced to $H_2S$ and S. Thus, there will be $H_2S$ in the brine which is stored in the bedrock [35,38]. The exchanges with $H_2S$ could lead to the enrichment of $^2H$ in water [39]. Therefore, it indicates that the mixed water exchanged $^2H$ isotopes with $H_2S$ in these two years. Additionally, the average deviations of each indicators were small compared to the "only PCA" method which does not consider the hydrogeochemical analysis (Table 3). Conclusively, the calculation deviations of the mixing ratios were small. It suggests that the PCA (based on hydrogeochemical analysis) and the ternary mixture model performed well for determining end-members and computing mixing ratios.

*4.4. Discussion: The Potential Water Flow Channel*

The potential water flow channels and the prevailing supply patterns can be inferred from the results of mixing ratios based on the tectonic and engineering geological conditions. The mixing ratio characteristics of different water sites in the same sublevel and the difference of mixing ratio between different sublevels were discussed in order to obtain a comprehensive analysis. The distribution and movement of groundwater are mainly controlled by fractures, so the high proportion of the water end-member in the mine water proves that the fractures between the water site and the end-member are developed.

The seepage water sites are mainly distributed in the north of F3 (prospecting line 1780) and in −375 m, −510 m, and −600 m sublevels. Thus, the distribution rule of mixing ratio in this area was discussed, and this area was divided into two areas (Area I and Area II) with the prospecting line 2260 (Figure 8). Figure 9 shows the characteristics of the mixing ratios of three sublevels (−375 m, −510 m, and −600 m sublevels) in the above two areas (375-I represents the average mixing ratio of the water sites located in −375 m sublevel and in Area I). In −375 m sublevel, the water sites located in Area I had a relatively low proportion of seawater and a high proportion of freshwater, while the water sites located in the Area II had a relatively high proportion of seawater. However, in −510 m sublevel, the water sites located in the Area I had a high proportion of seawater, especially the 510-2 water site. In −600 m sublevel, the distribution rule of mixing ratio was similar to that in −375 m sublevel. The water sites located in the Area I had a relatively low proportion of seawater while the difference

is that these water sites had high proportions of brine (in −375 m sublevel, the water sites had high proportion of freshwater).

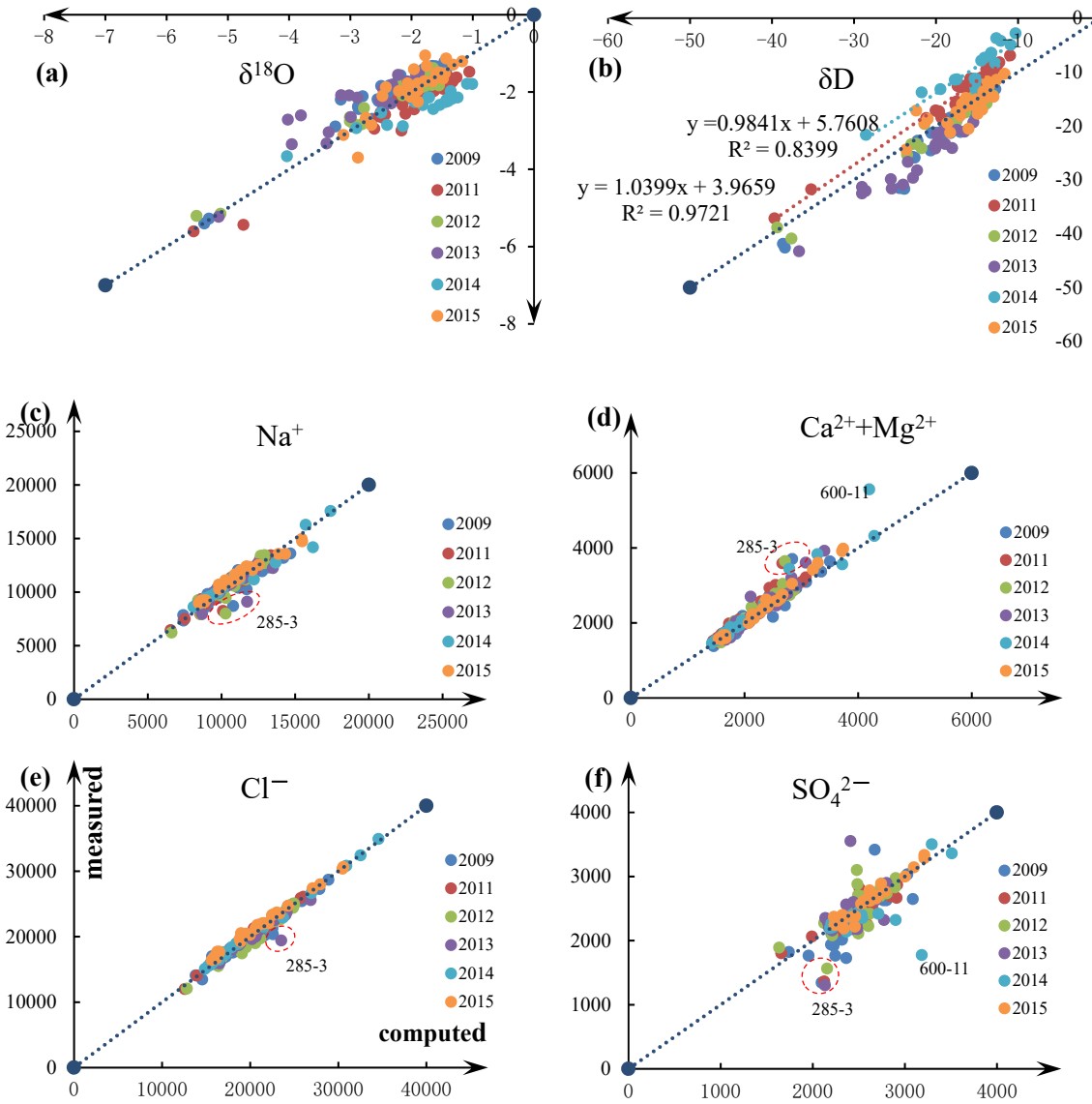

**Figure 7.** Relationship between computed concentrations (*x*-axis) versus measured concentrations (*y*-axis). Samples were collected from 2009 to 2015. ((**a**–**f**) is $\delta^{18}O$, $\delta D$, $Na^+$, $Ca^{2+} + Mg^{2+}$, $Cl^-$ and $SO4^{2-}$, respectively.)

The difference of seawater proportion between −510 m sublevel and −375 m and −600 m sublevels in Area I suggests that the prevailing supply pattern of seawater is lateral recharge. The land subsidence of the study area was monitored by GPS. Figure 8 shows the vertical displacement of the study area which is caused by settlement. The ground deformation suggests that a compressive zone is located at the center of the subsidence and a tensional zone is distributed around the compressive zone [40]. The horizontal fissures would be closed in the compressive zone because of the vertical stress. Therefore, some flow paths in this area will be blocked. While, in the tensional zone, the tensile stress makes the crack aperture become larger and new fissures are created. So, there will be more flow paths. The Area I was located in the compressive zone. Thus, the water sites in −375 m and −600 m sublevels had low proportions of seawater. However, the −510 m sublevel was different from these two sublevels and the difference is due to uneven subsidence. To some extent, the −420 m and −465 m sublevels interrupted

the upper subsidence because of the small exploitation quantity. That stopped the transmission of the upper pressure. What is more, the excavation in −510 m led to the stress releasing of the surrounding rock mass. Thus, the fissures would open and become the seepage channels of seawater. The Area II was located in the tensile zone. Thus, most water sites in this zone had a high proportion of seawater.

According to the tectonic conditions, the freshwater recharged the groundwater through the "Fracture Zone", so the supply pattern of freshwater was vertical recharge. In −465 m sublevel, the water sites 465-1 and 465-2, which are located in the Area I, had a high proportion of freshwater and a low proportion of seawater. It suggests that the main range affected by freshwater was −465 m sublevel and above in this area. In addition, some water sites around F3 also had a high proportion of freshwater, for example, 510-9 (the proportion of freshwater was 23%) and 510-10 (the proportion of freshwater was 24%) water samples. It shows that F3 is also the watercourse of freshwater.

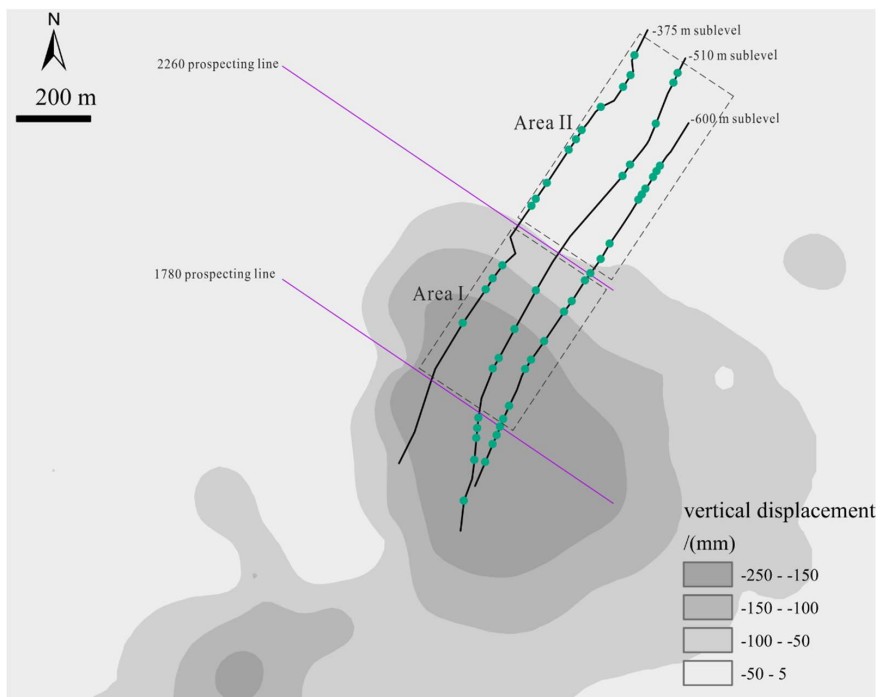

**Figure 8.** Vertical displacement caused by settlement in study area.

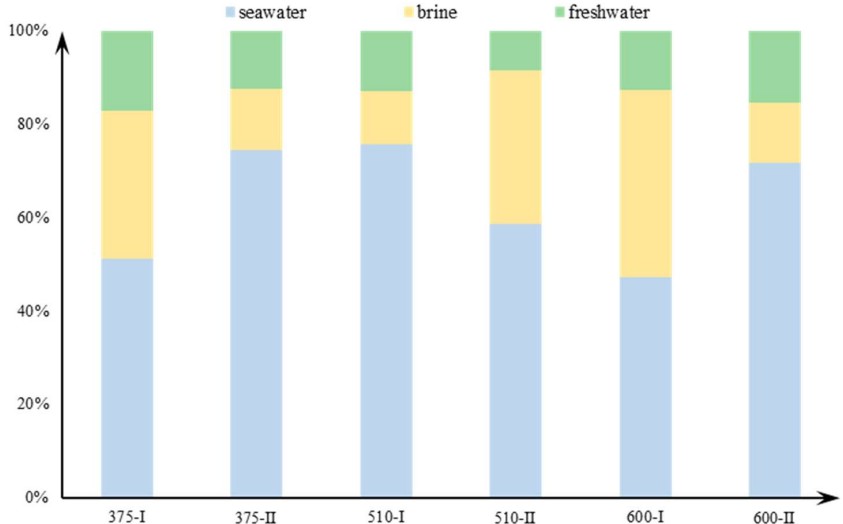

**Figure 9.** The mixing ratios of three sublevels in two areas.

## 5. Conclusions

The seawater intrusion and water in flow are potential disasters to a coastal mine. Once water inrush disaster occurred, it will bring great property damage and casualties to the mine. Monitoring and early identification of the origin of the mine water is helpful to provide support and guarantee the safety of deep coastal mining activities.

1 Non-conservative ion ($K^+$) and ion exchange ($Ca^{2+}$ and $Mg^{2+}$) were analyzed by using the hydrogeochemical analysis.

2 Three water end-members (seawater, freshwater, and brine) were identified by using the PCA based on the hydrogeochemical analysis.

3 The proportion of each water end-member were quantified by a ternary mixture model and the calculation results of the mixing ratios suggest that the proportion of seawater in mine water is about 55% and the freshwater is about 15% for the entire mine. The water sites located in Area II (a tensile zone) had a high proportion of seawater.

4 The potential water flow channels and the prevailing supply patterns were discussed by the results of mixing ratios based on the tectonic and engineering geological conditions. It shows that the prevailing supply pattern of seawater is lateral recharge in Area I and the uneven subsidence leads to the difference of seawater proportion between −510 m sublevel and −375 m and −600 m sublevels; The influence depth of freshwater was about −500 m; F3 was both a watercourse for seawater and fresh water.

**Author Contributions:** Conceptualization, X.D.; Data curation, H.G., S.L. and Q.S.; Funding acquisition, F.M.; Investigation, H.G., S.L. and Q.S.; Methodology, X.D.; Project administration, F.M.; Software, X.D.; Writing–original draft, X.D.; Writing–review & editing, F.M., J.G., H.Z. and H.G.

**Funding:** This research was funded by the National Science Foundation of China (Grant Nos. 41831293, 41907174), and National Key Research Projects of China (2016YFC0402802).

**Conflicts of Interest:** The authors declare no conflict of interest.

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
