# Peer review of "Source Identification and Quantification of Seepage Water in a Coastal Mine, in China"

_water, doi:10.3390/w11091862_

Round 1

Reviewer 1 Report

This paper presents a mixing model for mine waters in a coastal mining area. As such it is likely to be of interest to the readers of Water. The method and approach are interesting, although from the final PCA analysis the same result could have been achieved by plotting chloride or TDS against hydrogen isotope ratio, so the use of PCA in this instance is not that definitive. There are numerous areas of poor writing style throughout which I have highlighted on the attached manuscript. There are also some issues with the content or scientific terminology that require some work.

1) The definition of water end members includes brines, which are stated to be formed by evaporation of palaeo-seawater. No evidence is presented for this statement. There should either be a citation, or the origin of the brines should be included in the discussion section. A short section at the start of the discussion explaining the end members and their inferred origin might be useful.

2) Quaternary water is used as an end member. From reading the manuscript this refers to water hosted in the Quaternary sedimentary aquifers, which are the result of mixing between seawater and freshwater. This should explained, and a new term used. Quaternary water implies a Quaternary age palaeowater which is not the case.

3) The explanation of the hydrogeology is weak - reference to underground water courses need to clarified and the possible presence of matrix flow explained in more detail.  This also includes terms for hydraulic properties. In line 91-92 viscosity and impermeability of an aquifer are discussed. Presumably the authors mean porosity and permeability?

4) The statement of the significance of the findings at several points says that study of water chemistry can prevent water inflow. This is wrong. Interpretation of water chemistry can help inform models used to design preventative measures, but it can't directly prevent it.

5) Ca and Mg are interpreted as reflecting ion exchange. This is fine and is probably correct, but the authors should also briefly consider carbonate dissolution (i.e. plot Ca and Mg against HCO3).

6) Shifts in hydrogen isotope ratio are attributed to exchange with H2S. this needs more discussion as this aspect of the isotope data is not really covered at present.

7) Figure 8 needs far more explanation, particularly in the caption. From the text I think this shows a map of ground displacement but this is far from clear. the link between this and possible flow paths needs clarifying in the discussion.

Author Response

We would like to express our sincere thanks to the reviewer for the constructive and positive comments.

Replies to Reviewer 1

General Comments:

The method and approach are interesting, although from the final PCA analysis the same result could have been achieved by plotting chloride or TDS against hydrogen isotope ratio, so the use of PCA in this instance is not that definitive.

Answer: In this study, the water end-members could also be identified by plotting chloride or TDS against hydrogen isotope ratio. The second purpose of using PCA was to get more precise calculation results of mixing ratios. The principal components could account for more information of the water samples than using original variables (Cl or TDS against δ18O or δD). Therefore, using principal components is more representative than using some of original variables in the calculation of mixing ratios. Additional explanation has been added in the revised version. Hope you understand. (Line 52-54)

Specific Comments:

Firstly, thank you for pointing out the grammar and writing errors . Corrections have been made in the revised version.

The definition of water end members includes brines, which are stated to be formed by evaporation of paleo-seawater. No evidence is presented for this statement. There should either be a citation, or the origin of the brines should be included in the discussion section. A short section at the start of the discussion explaining the end members and their inferred origin might be useful.

Answer: Correction has been made in the revised version. We added a description of each water end-members at the start of this section and added three references to the origin of brine. (Line 159-161)

Quaternary water is used as an end member. From reading the manuscript this refers to water hosted in the Quaternary sedimentary aquifers, which are the result of mixing between seawater and freshwater. This should explained, and a new term used. Quaternary water implies a Quaternary age palaeo water which is not the case.

Answer: Thank you for pointing out the grammar and writing errors . In this study, Quaternary water refers to water hosted in the Quaternary aquifers while for the convenience of narration, the Quaternary aquifer water is simply referred to as Quaternary water.  We added some explanations for Quaternary water in “2.2 Hydrogeological Conditions”. (Line 131-138 and Line 156-159)

The explanation of the hydrogeology is weak - reference to underground water courses need to clarified and the possible presence of matrix flow explained in more detail. This also includes terms for hydraulic properties. In line 91-92 viscosity and impermeability of an aquifer are discussed. Presumably the authors mean porosity and permeability?

Answer: Correction has been made in the revised version. It was revised as “Therefore, in deep underground granite, fracture flow dominates while the matrix flow is mainly present in the Quaternary aquifer in the shallow ground.” (Line 71-72) In addition, we rewrote the “Hydrogeological Conditions” to make it more detailed and clear. (Line 128-150)

The statement of the significance of the findings at several points says that study of water chemistry can prevent water inflow. This is wrong. Interpretation of water chemistry can help inform models used to design preventative measures, but it can't directly prevent it.

Answer: Thank you for pointing out this mistake. It was revised as “…and can help inform models used to design preventative measures of water inrush…” (Line 83-85)

Ca and Mg are interpreted as reflecting ion exchange. This is fine and is probably correct, but the authors should also briefly consider carbonate dissolution (i.e. plot Ca and Mg against HCO3).

Answer: Thanks for your suggestion. We have added the plot of Ca and Mg against HCO3 in the revised version and consider the influence of carbonate dissolution. (Line 272-275)

Shifts in hydrogen isotope ratio are attributed to exchange with H2S. this needs more discussion as this aspect of the isotope data is not really covered at present.

Answer: Correction has been made in the revised version. We have added a more detailed explanation. It was revised as “Thus, there were D-shifts (the enrichment of D) in the water samples. When the space where water is stored is closed for a long time, desulfurization will occur, and SO42− will be reducted to H2S and S. Thus, there will be H2S in the brine which is stored in the bedrock [35, 38]. And the exchange with H2S could lead to the enrichment of 2H in water [39]. Therefore, it indicates that the mixed water exchanged H isotopes with H2S in these two years.” (Line 341-345)

Figure 8 needs far more explanation, particularly in the caption. From the text I think this shows a map of ground displacement but this is far from clear. the link between this and possible flow paths needs clarifying in the discussion.

Answer: Correction has been made in the revised version. We changed the caption to “Vertical displacement caused by settlement in study area” and changed the “sedimentation value” to “vertical displacement”. In addition, we added an explanation for the link between the figure and the flow paths. (Line 379-385)

Other comments in the attachment:

Line 19-21: “The results indicate that the proportion of seawater in mine water is about 55% and the freshwater is about 15% for the entire mine;” This needs more explanation - are there other components to take this up to 100%?

Answer: Correction has been made in the revised version. It was revised as “The results indicate that the proportion of seawater in mine water is about 57%, the freshwater is about 16% and the brine is about 27% for the entire mine area” (Line 20-21)

2. Line 23-24: “…“Fracture Zone” in a vertical recharge and the influence depth was about -500 m; F3…” "Fracture zone" and F3 need explaining in the abstract.?

Answer: Correction has been made in the revised version. We have described the “Fracture Zone” in another way and added an explanation to F3. (Line 24-26)

3. Line 68-69: “Thus, where there is water flowing, there must be a fissure (underground watercourse).” This is a poor summary of the hydrogeology of the situation. there will still be matrix flow, but fracture flow may dominate.

Answer: Thank you for pointing out this mistake. It was revised as “Therefore, in deep underground granite, fracture flow dominates while the matrix flow is mainly present in the Quaternary aquifer in the shallow ground. Thus, fissures (underground watercourse) can be inferred by the seepage water in the tunnel of the mine.”  (Line 71-74)

4. Line 121: “…does it destroy the aquifer of F1,” in what way does it destroy the aquifer? needs rephrasing.

Answer: Correction has been made in the revised version. It was revised as “F3 offsets the ore body and F1 by 10 to 20 m, which makes the aquifuge of F1 partially destroyed (the fault gouge of F1 is cut).” (Line 124-125)

5. Line 154: “…“ion (K+, Na+, Ca2+, Mg2+, Cl, SO42, HCO3, and NO3) concentrations…” why was Br not included as a conservative tracer?

Answer: Thank you, this is a good suggestion, we will consider it in the future work.

6. Line 167: “The PCA results are accurate on the premise…” robust? accurate refers to how close a measurement is to the true value and does not apply to PCA.

Answer: Thank you for pointing out this mistake. It has been corrected. (Line 192)

7. Line 259: “…are also conservative species which can be used to identify different water sources.” They can also be affected by water-rock interaction.

Answer: Correction has been made in the revised version. In “Introduction” we mentioned “Because of the high flow rate and low temperature of mine water, the chemical and isotopic reactions (ion exchange and water–rock interaction) have a little effect on it. Thus, the mine water is mainly dominated by the processes of mixing. The ion exchange reactions were identified by hydrogeochemical analysis and the isotopic reactions were analyzed in the deviation analysis of the mixing ratios.” (Line 76-80) So here we additional explanation. It was revised as “…are also relatively conservative species (the small part of isotopic reactions will be analyzed in the deviation analysis)” (Line 286-287)

8. Line 286-292: “The mixing ratio calculation results of all mine water are listed in Table 3. In general, the proportion of seawater in mine water was about 55% and…” Should state that the remainder is brine.

Answer: Thank you for pointing out this mistake. We have added the proportion of brine. (Line 312-313)

9. Line 396-397: “…but its 396 protection has not been used in practice” unclear, rephrase

Answer: Correction has been made in the revised version. We have deleted this sentence. (Line 405)

Reviewer 2 Report

The topic of the paper is certainly one of the interest to the hydrogeological and hydrochemical community as its objective is the determination of the water end-members of the seepage water in a mine and the quantification of the proportion of them in order the seawater intrusion and water inrush phenomena to be estimated. I found it very interesting and I really believe that it will contribute to the protection of mines from disasters caused by the seawater intrusion and water inrush phenomena. The results are clearly presented and the conclusions are fully supported by the results. But, the use of English language is problematic in some parts of the manuscript and this does not help the study to be clearly understood.

General Comment

The paper needs to be crosschecked to correct grammar/syntax errors. It must be reviewed by an efficient (preferably native) English-speaking reviewer before being published.

Specific Comments

Abstract

Authors should use dots at the end of the periods in the abstract.

Line 12: “Seawater intrusion and water inrush are potential disasters in the…”

Seawater intrusion and water inrush cannot be disasters! These are phenomena that cause disasters. Please rephrase

Lines 13-14: “……determine the water end-members of the seepage water…”

Authors should write down the end members as they did at the end of the manuscript.

Lines 16-17: “Finally, a ternary mixture model was applied to compute the mixing ratios”

I propose the authors to avoid the use of word “Finally” in the middle of the abstract. Furthermore it does not explained here why the ternary mixture model was chosen. A sentence must be added given the information that three end-members were selected.

Line 18: “…..and the possible supply patterns were inferred…..”

I don’t understand the word “possible”. The author should be more specific on this. Maybe authors refers the “prevailing supply patterns based on the results of the method that is used”? The way it has been written, authors tend to place the readers’ focus on the uncertainty of your calculations. Again, if authors clarify the characterization “possible” they will better present the scientific background of their research.

Lines 20-21: “…… for entire mine;”

It is preferable to add the word area, as I understand it à “…… for entire mine area;”

Line 21: “The mainly supply pattern…..”

Do the authors mean the most prevailing supply source? If yes, please rephrase.

Line 21: “In -510 m sublevel and Area Ⅱ….”

Someone could not understand which the Area II is by reading the abstract because it is explained in the manuscript. So, the authors should present this result in another way, by rephrasing the sentence.

Line 23: “…..mainly through the “Fracture Zone” in a vertical…..”

Fracture Zone is the name of the zone? If that zone has the characteristic of being fractured authors should skip the “…” so that the reader understands that the area is fractured and not just named as fractured. If the area is considered fractured authors should not mention it in abstract but in your main text body.

Line 24: “……F3 was both….”

Someone could not understand what the F3 is by reading the abstract because it is explained in the manuscript. So, the authors should present this result in another way, by rephrasing the sentence.

Introduction

Lines 32-33: “Thus, groundwater is the 32 potential threat to the mine….”

It is a bit vague. As I read it I understand that the presence of groundwater is a threat for the mining operations to take place. I believe that authors mean that the issue for the mine area is the presence of freshwater and seawater. Please, rephrase the sentence so as the proper meaning to be passed to the reader.

Line 47:“….used to analyze that means only a small 47 part of the water sample information are considered.”

To analyze what? Maybe the authors mean “to be analyzed”?

Line 59:“…so sometimes it would produce….”

Not “…it…” but “…they would produce….”

Lines 62-63: “…..and the selected species should be conservative…”

Why they should be conservative? Please mention the reason why. As I read I understand that the authors want to mention that that the “the selected species should be conservative so as to represent most of the information of the water sample”. If this is the meaning please rephrase. If not please explain it in a better way.

Line 72:  “….is mainly dominated by mixing and the reaction…”

Please add “the processes of” before the word “mixing” à “….is mainly dominated by the processes of mixing and the reaction…”

Line 73: “….and isotopic reaction…”

Please add “the” after the “and” because they are different methods.

Study area

Line 94. “Its thickness increase from north to south.”

Please rephrase à “Its thickness increases from north to south.”

Line 107: “…..there are some typical secondary… ”

Typical” is a vague expression for describing the degree of fracturing. Please rephrase.

Line 110: “…which is good for resisting water….”

Good what? Proportion? Value? Please be more specific.

Line 110: “This fault, forming before…”

“This fault, formed.”

Line 119: “Thus, F3 is good for water storage…”

Again here the use of word “good” is not the proper. Please be more specific by rephrasing.

Line 121: “…does it destroy the aquifer of F1,..”

The use of word destroy is kind of vague expression. In what sense does it destroy the aquifer?

Materials and methods

Line 147: “….the Wanghe River south of the mine”

Please add the word “located” as follow: “….the Wanghe River located south of the mine”

Line 148: “…in the Bohai Sea west…”

Please add the word “located” as follow: “…in the Bohai Sea located west…”

Line 199:“And only one species (species a was selected in Figure 3)….”

Please replace word “species” with “specie”.

Equation 4: The symbols of concentration are in small characters, while in line 213 are symbolized with Capita characters. Please use a uniform presentation.

Figure 4: The equations are on the spots and on each other. They are not easily seen.

Lines 229-230: “And the minimum and maximum of the K+ concentrations of mine water (a large floating range) show that the K+ is non-conservative species.”

As this information is very important, please rephrase the sentence without the use of parenthesis. For example: “The large floating range of the minimum and maximum values of the K+ concentrations of mine water indicates that……”

Line 270: “….and the mine water are obvious shown in the plot…”

Please rephrase the sentence.

Author Response

We would like to express our sincere thanks to the reviewer for the constructive and positive comments.

Replies to Reviewer 1

General Comments:

The paper needs to be crosschecked to correct grammar/syntax errors. It must be reviewed by an efficient (preferably native) English-speaking reviewer before being published.

Answer: Thank you for pointing out the mistake. Since we have modified most of the manuscript, including the “Introduction” and the “Study area”, we haven't had time to polish it. If it is not suitable, we will polish it as soon as possible. Hope you understand. 

Specific Comments:

Abstract

Line 12: “Seawater intrusion and water inrush are potential disasters in the…” Seawater intrusion and water inrush cannot be disasters! These are phenomena that cause disasters. Please rephrase.

Answer: Correction has been made in the revised version. It was revised as “The Sanshandao Gold Mine, which is the largest coastal mine in China, is under threat from seawater intrusion and water inrush.” (Line 12-13)

Line 13-14: “……determine the water end-members of the seepage water…” Authors should write down the end members as they did at the end of the manuscript.

Answer: Correction has been made in the revised version. (Line 14)

Line 16-17: “Finally, a ternary mixture model was applied to compute the mixing ratios” I propose the authors to avoid the use of word “Finally” in the middle of the abstract. Furthermore it does not explained here why the ternary mixture model was chosen. A sentence must be added given the information that three end-members were selected.

Answer: Thank you for pointing out this mistake. It was revised as “Three end-members were identified, so a ternary mixture model was applied to compute the mixing ratios.” (Line 17)

line 18: “…..and the possible supply patterns were inferred…..” I don’t understand the word “possible”. The author should be more specific on this. Maybe authors refers the “prevailing supply patterns based on the results of the method that is used”? The way it has been written, authors tend to place the readers’ focus on the uncertainty of your calculations. Again, if authors clarify the characterization “possible” they will better present the scientific background of their research.

Answer: Correction has been made in the revised version. It was revised as “…the prevailing supply patterns were inferred by…” (Line 18)

5. Line 20-21: “…… for entire mine;” It is preferable to add the word area, as I understand it à “…… for entire mine area;”

Answer: Thank you for pointing out this mistake. It has been corrected. (Line 21)

6. Line 21: “The mainly supply pattern…..” Do the authors mean the most prevailing supply source? If yes, please rephrase.

Answer: This means that the groundwater is supplied by the lateral recharge of seawater. So the ‘lateral recharge’ refers to the supply pattern of seawater. It was revised as “The prevailing supply pattern…” (Line 21)

7. Line 21: “In -510 m sublevel and Area Ⅱ….” Someone could not understand which the Area II is by reading the abstract because it is explained in the manuscript. So, the authors should present this result in another way, by rephrasing the sentence.

Answer: Correction has been made in the revised version. It was revised as “The water samples which were located in -510 m sublevel or in the northeast of prospecting line 2260 had high proportions of seawater” (Line 22-23)

8. Line 23: “…..mainly through the “Fracture Zone” in a vertical…..” Fracture Zone is the name of the zone? If that zone has the characteristic of being fractured authors should skip the “…” so that the reader understands that the area is fractured and not just named as fractured. If the area is considered fractured authors should not mention it in abstract but in your main text body.

Answer: Thank you for your suggestion. It was revised as “…mainly through the secondary fractures developed area in a vertical …” (Line 24)

9. Line 24: “……F3 was both….” Someone could not understand what the F3 is by reading the abstract because it is explained in the manuscript. So, the authors should present this result in another way, by rephrasing the sentence.

Answer: Correction has been made in the revised version. We added the explanation about F3 to make it clear. (Line 25-26)

Introduction

Lines 32-33: “Thus, groundwater is the potential threat to the mine….” It is a bit vague. As I read it I understand that the presence of groundwater is a threat for the mining operations to take place. I believe that authors mean that the issue for the mine area is the presence of freshwater and seawater. Please, rephrase the sentence so as the proper meaning to be passed to the reader.

Answer: Thank you for pointing out this mistake. It has been corrected. (Line 34)

Line 47: “….used to analyze that means only a small part of the water sample information are considered.” To analyze what? Maybe the authors mean “to be analyzed”?

Answer: Correction has been made in the revised version. It was revised as “…used for analysis which means only a small part of the water sample information are considered.” (Line 47)

Line 59: “…so sometimes it would produce….” Not “…it…” but “…they would produce….”

Answer: Thank you for pointing out this mistake. It has been corrected. (Line 61)

Lines 62-63: “…..and the selected species should be conservative…” Why they should be conservative? Please mention the reason why. As I read I understand that the authors want to mention that that the “the selected species should be conservative so as to represent most of the information of the water sample”. If this is the meaning please rephrase. If not please explain it in a better way.

Answer: Correction has been made in the revised version. It was revised as “…the selected species should be conservative so as to get precise calculation results. However, usually one or two chemical and isotopic species can not represent most of the information of the water sample.” (Line 64-66)

Line 72: “….is mainly dominated by mixing and the reaction…” Please add “the processes of” before the word “mixing” à “….is mainly dominated by the processes of mixing and the reaction…”

Answer: Thank you for pointing out this mistake. It has been corrected. (Line 78)

Line 73: “….and isotopic reaction…” Please add “the” after the “and” because they are different methods.

Answer: Thank you for pointing out this mistake. It has been corrected. (Line 79)

Study area

Line 94. “Its thickness increase from north to south.” Please rephrase à “Its thickness increases from north to south.”

Answer: Thank you for pointing out this mistake. It has been corrected. (Line 95)

Line 107: “…..there are some typical secondary… ” Typical” is a vague expression for describing the degree of fracturing. Please rephrase.

Answer: Correction has been made in the revised version. We deleted the word ‘typical’, because there was a detailed description of ‘secondary fractures’ at the end of the section. (Line 148-150)

Line 110: “…which is good for resisting water….” Good what? Proportion? Value? Please be more specific.

Answer: Correction has been made in the revised version. It was revised as “Under normal conditions, the fault gouge is tightly squeezed, which excludes water.” (Line 113-114)

Line 110: “This fault, forming before…”“This fault, formed.”

Answer: Thank you for pointing out this mistake. It has been corrected. (Line 108)

Line 119: “Thus, F3 is good for water storage…” Again here the use of word “good” is not the proper. Please be more specific by rephrasing.

Answer: Correction has been made in the revised version. It was revised as “This makes the rock in the fracture zone extremely fractured, which provides space for the storage and migration of groundwater.” (Line 121-123)

Line 121: “…does it destroy the aquifer of F1,..” The use of word destroy is kind of vague expression. In what sense does it destroy the aquifer?

Answer: Correction has been made in the revised version. It was revised as “F3 offsets the ore body and F1 by 10 to 20 m, which makes the aquifuge of F1 partially destroyed (the fault gouge of F1 is cut).” (Line 124-125)

Materials and methods

Line 147: “….the Wanghe River south of the mine” Please add the word “located” as follow: “….the Wanghe River located south of the mine”

Answer: Thank you for pointing out this mistake. It has been corrected. (Line 163)

Line 148: “…in the Bohai Sea west…” Please add the word “located” as follow: “…in the Bohai Sea located west…”

Answer: Correction has been made in the revised version. (Line 162)

Line 199:“And only one species (species a was selected in Figure 3)….” Please replace word “species” with “specie”.

Answer: Thank you for pointing out this mistake. It has been corrected. (Line 223)

Equation 4: The symbols of concentration are in small characters, while in line 213 are symbolized with Capita characters. Please use a uniform presentation.

Answer: Thank you for pointing out this mistake. It has been corrected. (Line 237)

Figure 4: The equations are on the spots and on each other. They are not easily seen.

Answer: Thanks for your suggestion. We re-adjusted the figure to make it clearer. (Line 280-282)

Lines 229-230: “And the minimum and maximum of the K+ concentrations of mine water (a large floating range) show that the K+ is non-conservative species.” As this information is very important, please rephrase the sentence without the use of parenthesis. For example: “The large floating range of the minimum and maximum values of the K+ concentrations of mine water indicates that……”

Answer: Correction has been made in the revised version. It was revised as “The large floating range of the minimum and maximum values of the K+ concentrations of  mine water indicates that the K+ is non-conservative species.” (Line 253-254)

Line 270: “….and the mine water are obvious shown in the plot…” Please rephrase the sentence.

Answer: Correction has been made in the revised version. It was revised as “The characteristics of all water samples (including four water sources and all mine waters) are obvious shown in the plot…” (Line 295)
